# Pangenomic and Phenotypic Characterization of Colombian *Capsicum* Germplasm Reveals the Genetic Basis of Fruit Quality Traits

**DOI:** 10.3390/ijms26178205

**Published:** 2025-08-23

**Authors:** Maira A. Vega-Muñoz, Felipe López-Hernández, Andrés J. Cortés, Federico Roda, Esteban Castaño, Guillermo Montoya, Juan Camilo Henao-Rojas

**Affiliations:** 1Departamento de Ciencias Farmacéuticas, Biomédicas y Veterinarias, Facultad Barberi de Ingeniería, Diseño y Ciencias Aplicadas, Universidad Icesi, Cali 760031, Colombia; maira.vega@u.icesi.edu.co (M.A.V.-M.); esteban.castano@u.icesi.edu.co (E.C.); 2Corporación Colombiana de Investigación Agropecuaria (AGROSAVIA)—C.I. La Selva, Km 7 vía Las Palmas, Rionegro 054048, Colombia; llopez@agrosavia.co (F.L.-H.); acortes@agrosavia.co (A.J.C.); 3Facultad de Ciencias Agrarias—Departamento de Ciencias Forestales, Universidad Nacional de Colombia—Sede Medellín, Medellín 050034, Colombia; 4Max Planck Tandem Group GEME, Facultad de Ciencias Agrarias, Universidad Nacional de Colombia, Bogotá D.C. 111321, Colombia; frodaf@unal.edu.co; 5BioInc. Center for Biomass Valorization Research, Universidad Icesi, Cali 760031, Colombia; 6Grupo de Investigación en Sustancias Bioactivas, Facultad de Ciencias Farmacéuticas y Alimentarias, Universidad de Antioquia UdeA, Calle 70 No. 52-21, Medellín 050010, Colombia

**Keywords:** genotyping-by-sequencing (GBS), GWAS, polygenic architecture, northwest South America, capsaicinoids

## Abstract

*Capsicum* is one of the most economically significant vegetable crops worldwide, owing to its high content of bioactive compounds with nutritional, pharmacological, and industrial relevance. However, research has focused on *C. annuum*, often disregarding local diversity and secondary gene pools, which may contain hidden variation for quality traits. Therefore, this study evaluated the genetic and phenotypic diversity of 283 accessions from the Colombian germplasm collection in the agrobiodiversity hotspot of northwest South America, representing all five domesticated species of the genus. A total of 18 morphological, physicochemical, and biochemical fruit traits were assessed, including texture, color, capsaicinoid, and carotenoid content. The phenotypic data were integrated with genomic information obtained through genotyping-by-sequencing (GBS) using the *C. annuum* reference genome and a multispecies pangenome. Fixed-and-Random-Model-Circulating-Probability-Unification (FarmCPU) and Bayesian-information-and-Linkage-disequilibrium-Iteratively-Nested-Keyway (BLINK) genome-wide association studies (GWAS) were performed on both alignments, respectively, leading to the identification of complex polygenic architectures with 144 and 150 single nucleotide polymorphisms (SNPs) significantly associated with key fruit quality traits. Candidate genes involved in capsaicinoid biosynthesis were identified within associated genomic regions, terpenoid and sterol pathways, and cell wall modifiers. These findings highlight the potential of integrating pangenomic resources with multi-omics approaches to accelerate *Capsicum* improvement programs and facilitate the development of cultivars with enhanced quality traits and increased agro-industrial value.

## 1. Introduction

The genus *Capsicum* ranks among the most economically important crops within the Solanaceae family. Between 2013 and 2023, the production of both chili and sweet peppers grew substantially, with dried forms increasing by 63.0% and fresh forms increasing by 22.4% [1]. This growth reflects rising demand for functional, nutrient-rich foods and the expanding use of *Capsicum* species in nutraceutical, pharmaceutical, and industrial applications due to their high content of bioactive compounds [2,3]. Advances in chemical technologies have further supported research on *Capsicum* species metabolites and their by-products, facilitating alignment with evolving market demands [4]. The most relevant specialized metabolites in this genus include capsaicinoids (pungency), carotenoids (color and flavor, e.g., capsanthin, capsorubin, lutein, α- and β-carotene), and phenolic compounds (antioxidant activity). *Capsicum* fruits are also rich in vitamins (provitamin A, C, E), minerals (Fe, Ca, Mg, K), proteins, carbohydrates, fatty acids, and terpenes. These components have been associated with potential analgesic, anti-inflammatory, antitumor, anti-obesity, antidiabetic, cardiovascular, and dermatological effects [2,5,6].

The *Capsicum* genus includes approximately 43 accepted species [7], 5 of which are domesticated: *C. annuum*, *C. chinense*, *C. frutescens*, *C. baccatum*, and *C. pubescens* [8]. Native to the temperate, subtropical, and tropical regions of the Americas, the Andes Mountains represent a major center of diversity [9]. The genus exhibits extensive genetic variability, comprising over 50,000 known varieties with significant morphological and metabolic heterogeneity [10,11]. Colombia, located within the northern Andean diversity hotspot, a mandatory confluence region for the Mesoamerican and South American floras, conserves a large portion of this genetic richness in its national germplasm bank [12], likely harboring unique variation boosted by introgression and sharpened ecological variation [13].

Diversity in fruit morphology and biochemical profiles within this germplasm presents a unique opportunity to investigate and utilize genomic regions associated with trait diversification [14]. Understanding this variation enables the selection of genotypes for improved morphological, biochemical, and stress-resistance traits. Uncovering the genetic basis of valuable traits requires integrating high-resolution phenotyping and genotyping with genome-wide association analyses (GWAS). This strategy highlights useful markers to enhance breeding efficiency and candidate genes for biotechnological applications [15].

In recent years, several studies have focused on identifying genomic regions associated [16] with fruit quality traits in *Capsicum* using GWAS and QTL approaches. For instance, Liu et al. [17] constructed a graph-based pangenome from 500 wild and cultivated accessions, revealing distinct domestication sweeps between *C. annuum* and *C. baccatum*, as well as introgressions in *C. chinense* and *C. frutescens*. Likewise, López-Moreno et al. [18] and Fu et al. [19] uncovered multiple QTL and SNPs associated with key agronomic traits, confirming the complex and pleiotropic architecture of fruit domestication. Complementary studies have identified high-effect loci controlling capsaicinoid content [20,21], carotenoid pathways [22,23], primary metabolites [24], and fruit morphology [22], highlighting promising candidate genes such as *RGLG1*, *HDG11*, *ARF23*, and a UDP-glycosyltransferase. These findings collectively reinforce the utility of high-resolution genotyping in uncovering the complex genetic basis of fruit quality in *Capsicum*.

Furthermore, in the era of plant genomics, the use of multispecies reference genomes (pangenomes) enables comprehensive identification of genomic regions that are structurally diverse and absent from conventional linear references. These genomic resources facilitate deeper exploration and more effective utilization of genetic diversity within germplasm collections [16]. However, despite advances in sequencing technologies and the availability of reference genomes for various *Capsicum* species, most genomic knowledge remains centered on *C. annuum* [17].

Due to this, it continues being a major challenge to understand and compare the molecular evolution, the drivers of population structure and gene flow, and the selection footprints across wild and domesticated *Capsicum* species. This limited scope constrains the broader utilization of valuable genetic resources conserved in germplasm banks. In order to strengthen commercial applications and breeding strategies aimed at enhancing fruit quality and industrial value, it is essential to reconstruct the genomic architecture of key traits across diverse *Capsicum* germplasm, spanning hidden hotspots of agrobiodiversity as in northwest South American. Therefore, this study aimed leveraging a multi-omics framework for the *Capsicum* germplasm in Colombia, integrating through GWAS phenotypic characterizations at the fruit level with SNP data derived from both reference and pangenomic assemblies. This way, by identifying key genomic regions and candidate genes associated with fruit quality traits, this study provides a robust foundation for targeted breeding and valorization of *Capsicum* resources.

## 2. Results

### 2.1. Phenotypic Traits

The evaluated *Capsicum* germplasm exhibited wide phenotypic diversity in fruit morphology, including various shapes, sizes, and colors (Figure 1). This variability was quantitatively supported by substantial differences among accessions in traits such as fruit width (0.53–10.77 cm; variance = 1.84), length (0.97–19.23 cm; variance = 9.87), weight (0.3–127.36 g; variance = 457.43), and number of seeds per fruit (0–295; variance = 1494.94). Likewise, the metabolite composition and physicochemical properties varied markedly among accessions, consistent with the findings reported by Castaño et al. [14]. For comparability with commercially processed products, fruit pastes from seeds, pericarp, and placenta were analyzed across multiple traits.

Measured parameters included pH, total and soluble solids, and texture attributes (firmness, consistency, cohesiveness, and work of cohesion). Colorimetric data encompassed lightness (L), chromatic coordinates (a and b), chroma (C), and hue angle (h°). Additionally, capsaicinoids (capsaicin, dihydrocapsaicin), total carotenoids and morphological descriptors (seed number, fruit weight, length, and width) were quantified. Correlation analysis revealed strong and significant associations among several expected trait pairs. High correlations were found among colorimetric parameters (r = 0.63–0.87), textural attributes (r = 0.70–0.99), morphological traits (r = 0.42–0.84), and chemical components (r = 0.80–0.97). Particularly strong correlations were observed between capsaicin and dihydrocapsaicin, as well as between total and soluble solids. In contrast, pH and total and soluble solids showed no significant associations with other traits (r < 0.50), suggesting a more independent behavior. Moreover, no significant correlations were observed between chemical and textural parameters, or between chemical and colorimetric variables with pH and total and soluble solids (Appendix A). This lack of association may be attributed to the broad genetic diversity within the population, reflecting high inter- and intra-specific variability.

### 2.2. Genotypic Data

Library construction for genotyping-by-sequencing (GBS) yielded libraries with a mean insert size of 323.02 bp (SD = 32.58) and an average concentration of 173.62 nM (SD = 91.97), as detailed in Appendix A. All libraries satisfied the minimum sequencing depth threshold of 5X, ensuring adequate coverage for downstream variant discovery. Quality metrics indicated high consistency across samples, with an average duplication rate of 35.59% (SD = 16.10%), a mean GC content of 41.09% (SD = 3.45%), and an average of 3,403,533 (SD = 1,807,881) sequencing reads per sample.

The resulting reads from GBS were aligned to two genomic assemblies: the *C. annuum* reference genome (GCA_002878395.3) [25] and a pangenome assembled from *C. annuum* var. *annuum* Zhangshugang, *C. baccatum* var. *pendulum* PI 632928, and *C. pubescens* Grif 1614 [17]. Starting from an initial set of 283 accessions, filtering steps retained 68,481 SNPs across 235 accessions in the reference-based dataset and 27,906 SNPs across 244 accessions in the pangenome-based dataset.

### 2.3. Population Structure and Genetic Clustering

Unsupervised PCA and *k*-means clustering analyses revealed five optimal genetic clusters within the dataset (*K* = 5), corresponding to the five domesticated species (Figure 2 and Appendix A). Notably, overlapping genetic profiles among *C. chinense*, *C. frutescens*, and *C. annuum* indicated shared ancestry and historical introgression, consistent with their classification within the *annuum* complex. Ancestry matrices derived from sNMF confirmed the presence of five ancestral populations, supporting the clustering results (Appendix A).

### 2.4. Trait-Associated SNPs (QTNs)

FarmCPU (Fixed and Random Model Circulating Probability Unification) and BLINK (Bayesian-information and Linkage-disequilibrium Iteratively Nested Keyway)-based GWAS models identified 144 significant SNPs in the reference genome associated with 16 fruit quality traits and 150 in the pangenome for all 18 evaluated traits (Figure 3, Figure 4 and Figure 5 and Appendix A). These algorithms enhance QTN detection by controlling population structure and false positives; FarmCPU alternates fixed and random models, while BLINK employs a faster, fixed-model approach based on Bayesian criteria [26,27]. In the reference genome, significant associations were detected for 4 morphological, 9 physicochemical, and 3 chemical traits. Notably, chromosomes 10, 2, 7, 3, and 12 harbored the highest number of trait-associated QTNs. In contrast, in the pangenome, QTNs were concentrated on chromosomes 12, 6, 5, 1, and 2 (Figure 6).

The range of SNP effect sizes revealed distinct patterns of genetic architecture across traits. In the reference genome, traits such as cohesion (–45.90 to –45.10) and capsaicinoids (–1.63 to 0.66) exhibited narrow effect intervals, reinforcing the contribution of small-effect variants. Similarly, in the pangenome, traits such as pH (0.23 to 0.23) and dihydrocapsaicin (–0.14 to 0.25) also displayed limited effect ranges, suggesting a polygenic basis predominantly governed by minor additive loci.

In contrast, traits such as consistency (–1999.42 to 738.94), firmness (–1352.69 to 286.86), and seed number per fruit (–38.57 to 50.26) in the reference genome, as well as consistency (± 1878.95), fruit firmness (255.08 to 955.99), and fruit weight (–99.94 to 166.69) in the pangenome, showed much broader effect ranges. These results point to the involvement of large-effect loci, consistent with more complex or potentially oligogenic architectures.

Regarding the number of associations, traits like cohesion (in the reference genome), and pH and Brix (in the pangenome) exhibited the fewest associated SNPs (2 each). In contrast, traits such as fruit weight (16 SNPs), capsaicin (14 SNPs), and soluble solids (13 SNPs) in the reference genome, as well as consistency (21 SNPs), luminosity (19 SNPs), and capsaicin content (13 SNPs) in the pangenome, displayed broader genetic architectures with associations distributed across multiple chromosomes. These findings reinforce the value of leveraging a pangenome to capture additional allelic variation and more accurately uncover the complex genetic basis of key agronomic and quality-related traits in *Capsicum.*

### 2.5. Candidate Gene Identification

A total of 434 and 425 candidate genes were, respectively, identified in the reference genome and the pangenome alignments within a 50 kb window around significant SNPs (Appendix A). Specifically, genes implicated in capsaicinoid biosynthesis (Figure 7) were identified through orthology analysis (Appendix A). These included 4CL at chromosome 1 (near position 25,405,966), BCKDH at chromosome 1 (near position 84,315,392), BCCP at chromosome 5 (near positions 220,528,081 and 252,793,727), and FatB at chromosome 3 (near position 34,056,297), all located near QTNs associated with capsaicin and dihydrocapsaicin content. Functional enrichment revealed nine additional candidate genes involved in phenylpropanoid, fatty acid, and terpenoid pathways, as well as those related to cell wall metabolism. Among these were CCL8 (at chromosome 3), 5-epiaristolochene synthase (at chromosome 12), cycloartenol synthase (at chromosome 4), galacturonosyltransferase 12 (at chromosome 3, and fasciclin-like arabinogalactan proteins (at chromosome 9).

### 2.6. Comparative Analysis of Reference Genome and Pangenome

Three QTNs were found in orthologous regions across both genomic assemblies. These included loci associated with capsaicin/capsaicinoid levels (chromosome 4), lightness (chromosome 6), and seed number per fruit (chromosome 1). The low level of overlap highlights the distinctiveness of each assembly and emphasizes the complementary value of using both the reference genome and the pangenome to capture a broader and more complete set of trait-associated loci.

## 3. Discussion

### 3.1. Rampant Introgression Reshapes Capsicum Diversity in Northwest South America

In terms of the demographic patterns that have shaped *Capsicum* diversity in northwest South America, supervised and unsupervised clustering analyses, based on genetic data from 235 and 244 accessions, respectively, for the reference genome and the pangenome alignments, revealed population stratification into five groups corresponding to the taxonomic classification of the five domesticated species captured in the germplasm bank. However, we found a closer genetic relationship between *C. chinense*, *C. frutescens*, and to a lesser extent, *C. annuum*, as evidenced by accessions clustering outside their assigned species. This genetic admixture is expected as per their classification within the “*annuum* complex”, a group of closely related species that evolved together in the South American lowlands. Their reproductive compatibility has facilitated historical hybridization and introgression events, blurring clear genetic differences, increasing heterozygosity, and making taxonomic boundaries more porous, sometimes resulting in intermediate phenotypes [30,31].

In the pangenome alignment, the differences in clustering and taxonomic classification are more evident. This is due to the pangenome incorporating a broader representation of genomic variation, including more inter-specific structural variants by simultaneously considering three reference genomes from different species [16]. Moreover, the ancestry matrix shows the presence of five ancestral populations, reinforcing the hypothesis of introgression among species. Thus, the *Capsicum* accessions from northwest South America appear as products of varying levels of genetic admixture, with assignment to a dominant ancestor consistent with their taxonomic classification. This indicates the presence of considerable genetic diversity influenced by components from other demographic groups, particularly evident in *C. chinense* and *C. frutescens*, in accordance with their shared evolutionary history [31].

### 3.2. Complex Polygenic and Pleiotropic Genomic Architectures Underly Fruit Quality Traits

Results from the BLINK and FarmCPU GWAS models are concordant between the reference genome and the pangenome in terms of the number of significant SNPs detected, supporting the robustness of the methods applied in this study. Although absolute numbers differed slightly between models and assemblies (56–88 and 68–82 QTNs in the reference genome and in the pangenome, respectively), both approaches identified relevant associations with fruit quality traits at the morphological, physicochemical, and chemical levels.

The findings also revealed a complex polygenic genomic architecture with a non-uniform distribution of associated SNPs across chromosomes. In the reference genome, chromosomes 10, 2, 7, 3, and 12 concentrate the most significant SNPs, whereas in the pangenome, chromosomes 12, 6, 5, 1, and 2 stand out. This variability in localization suggests that both assemblies offer complementary perspectives for identifying loci of interest. Conversely, certain chromosomes such as 1, 5, and 8 in the reference genome and 9, 7, and 10 in the pangenome show fewer associations, possibly due to simpler genetic architecture or limited contribution to the examined traits. Together, these trends highlight the polygenic nature of the analyzed quantitative fruit quality traits, so that many different genomic regions contribute small effects to a given trait, potentially influenced by dynamic environmental factors [32,33]. Ultimately, these results also reiterate the importance of integrating different genomic reference assemblies, specifically a reference genome approach vs. a pangenomic graph, to maximize the detection of functional variants for *Capsicum* breeding programs.

In terms of pleiotropy, the association analyses not only reveal multiple QTNs for a single trait (i.e., fulfilling the polygenic definition), but also QTNs significantly associated with multiple traits. This configuration is common in quantitative traits and has been reported in other crops [34,35,36,37]. It may reflect pleiotropic effects (one gene affecting multiple traits) or genetic correlations among similar traits [38].

In studies of *Capsicum* and other crops, pleiotropic loci have been identified that simultaneously modulate fruit quality, size, and pungency [15]. Thus, identifying QTNs for marker-assisted selection can improve multiple traits simultaneously. However, such a selection may also result in undesirable trade-offs, improving one trait while negatively affecting another. Therefore, it is necessary to study genetic correlations among traits and determine whether multiple associations from QTNs arise from pleiotropy or linkage, to adopt appropriate selection strategies such as genetic correlation breakers. In addition, epigenetic modifications may influence pleiotropy in manners that are yet to be further clarified [39].

### 3.3. Candidate Genes Guide Introgression Breeding of Fruit Quality Traits

Functional identification of candidate genes through orthologous relationships enables the linkage of genetic variants to biological functions relevant to specific phenotypic traits in *Capsicum*. We demonstrate how this approach allows specific functions to be assigned to genes associated with traits such as brightness and the production of secondary metabolites like capsaicinoids, compounds of both agronomic and commercial importance.

The most notable pathway retrieved is the capsaicinoid biosynthesis, including specialized secondary metabolites unique to this genus. Approximately 23 capsaicinoids have been reported [40], with capsaicin and dihydrocapsaicin accounting for 90% of capsaicinoids in the fruit [2]. Their biosynthesis involves the phenylpropanoid and branched-chain fatty acid pathways. Starting with phenylalanine, the phenylpropanoid pathway yields vanillylamine, while the branched-chain fatty acid pathway produces 8-methyl-6-nonenoic acid from valine [40].

Specifically, we identify the gene encoding the enzyme 4-coumaroyl-CoA ligase (4CL; XM_016716205). This enzyme catalyzes the conversion of p-coumaric acid to p-coumaroyl-CoA, an intermediate in the biosynthesis of vanillylamine, which serves as the direct precursor for the majority of capsaicinoids [28]. Moreover, we also detect the gene encoding the E1β subunit of 2-oxoisovalerate dehydrogenase (BCKDH-E1β; XM_047396937). This enzyme is responsible for the multistep oxidative decarboxylation of α-keto acids derived from branched-chain amino acids such as valine and leucine. Particularly, BCKDH-E1β catalyzes the synthesis of isobutyryl-CoA from α-ketoisovalerate, an intermediate that subsequently enters the branched-chain fatty acid biosynthetic pathway and ultimately contributes to capsaicinoid formation [29].

Likewise, within the same fatty acid biosynthesis pathway, we also capture the candidate gene encoding the biotin carboxyl carrier protein (BCCP; NM_001325012) of the acetyl-CoA carboxylase complex. BCCP constitutes a subunit of the heteromeric acetyl-CoA carboxylase enzyme, wherein biotin carboxylase first catalyzes the carboxylation of the carrier protein, and then the carboxyltransferase transfers the carboxyl group to acetyl-CoA, yielding malonyl-CoA [41,42]. In addition to the gene encoding the biotin carboxyl carrier protein, we also pinpoint the candidate gene FatB (XM_016708605), which encodes a palmitoyl-acyl carrier protein thioesterase. This thioesterase hydrolyzes saturated acyl-ACPs, thereby terminating the elongation of fatty acid chains during de novo biosynthesis and facilitating the production of 8-methyl-6-nonenoic acid, the immediate precursor of 8-methyl-6-nonenoyl-CoA [42,43,44].

Functional enrichment analysis reveals additional genes potentially regulating capsaicinoid variation within the studied population, particularly those involved in key metabolic pathways such as fatty acid, phenylpropanoid, and terpenoid biosynthesis. Beyond the previously validated genes, we uncover novel candidates not associated with capsaicinoid content. Among these is 4-coumarate-CoA ligase CCL8 (XM_016707209), which participates in both phenylpropanoid and fatty acid biosynthesis and exhibits medium-chain acyl-CoA ligase activity [28].

Furthermore, we capture two genes implicated in terpenoid and sterol production. First, CAS1 (XM_016714054) encodes a cycloartenol synthase that catalyzes the cyclization of 2,3-oxidosqualene to cycloartenol, representing the initial cyclic product in plant sterol biosynthesis [45,46]. Sterols are cyclic isoprenoid lipids crucial for maintaining membrane fluidity and flexibility [47]. Second, EAS12 (XM_047405883) encodes a 5-epiaristolochene synthase, which converts trans-farnesyl diphosphate into 5-epiaristolochene, an intermediate in the capsidiol biosynthetic pathway; subsequent hydroxylation steps yield capsidiol, a bicyclic sesquiterpenoid phytoalexin documented in *Capsicum annuum* as a defensive response to pathogen attack [48,49].

Conversely, capsaicinoids accumulate within the glandular epidermal cells of the interlocular septum in the fruit [50,51]. These metabolites are secreted by the glandular cells into subcuticular cavities—globular structures external to the cell wall—that expand into blister-like protrusions on the epidermis; this enlargement correlates with capsaicinoid accumulation [52,53]. Considering this, we find QTNs to be overrepresented in genes governing cell wall formation and glycosyltransferase activity.

Notably, we highlight two genes encoding fasciclin-like arabinogalactan protein 12 (FAS12; XM_016685978, XM_016685979). FAS12 is a glycoprotein featuring a fasciclin domain that localizes to the cell surface and plays essential roles in cell adhesion, as well as in the integrity and elasticity of the plant cell wall matrix; it is also proposed to facilitate protein–protein interactions [54,55]. Moreover, the gene GAUT12 (XM_016689675) encodes a galacturonosyltransferase 12 enzyme that is indispensable for homogalacturonan (HGA) biosynthesis, a pectic component constituting up to 60% of the pectin in primary cell walls, and additionally contributes to glucuronoxylan assembly in secondary cell walls [56,57,58]. Lastly, we identify three isoforms of α-1,3-arabinosyltransferase XAT2-3 (XM_016712030, XM_016711170, XM_016711169). These glycosyltransferases catalyze the arabinosylation of xylan, a principal hemicellulose component, by transferring arabinose residues to the xylan backbone to form branched arabinoxylans, which are integral to the structural integrity of the plant cell wall [59,60].

Identifying these candidate genes linked to the evaluated phenotypic traits represents a significant milestone in understanding the genetic architecture of *Capsicum* despite complex polygenic genomic architectures. Overall, these findings emphasize the importance of integrating comparative genomics with functional analysis to characterize candidate loci, especially in crops with high intraspecific diversity like *Capsicum*. Identifying candidate genes associated with specialized metabolites and fruit quality traits provides a solid molecular foundation for future gene-editing studies aimed at improving specific characteristics.

Finally, limitations in identifying genes that modulate differential traits considering both environmental and human benefits lie in their polygenic nature, often involving multiple metabolic pathways. This complexity highlights the need for a deeper understanding of the genetic, biochemical, and regulatory systems that control traits of interest [32]. Moreover, functional validation of significantly associated genes is essential through genetic complementation and genome editing techniques, along with replication of associations in independent populations [61].

### 3.4. Untapping Genomic Resources for Novel Breeding Targets

The *Capsicum* genus has been extensively studied using GWAS analyses, with a particular focus on *C. annuum*, leading to the identification of candidate genes associated with key phenotypic traits [15,30,62,63,64]. However, significant knowledge gaps remain, especially regarding the combined use of reference genomes and pangenomes to study morphological, physicochemical, and chemical fruit traits in other domesticated species across underutilized germplasm [65], such as the one found in the mega agrobiodiverse region of northwest South America. As the center of origin and diversity for this genus, the Americas harbor native varieties with highly valuable traits that surpass many commercial cultivars [9]. Therefore, the novelty of this study lays in integrating phenotypic and genomic characterizations from local unexplored germplasm to (i) unravel a more comprehensive picture of the underlying genetic architecture of fruit quality traits and to (ii) identify exotic alleles with potential applications in breeding programs aimed at novel agro-industrial demands. This would enable the identification and utilization of candidate genebank donors for developing improved cultivars [66].

Domestication and intense selection in modern commercial cultivars have favored specific morphological and metabolic traits such as high yield and fast growth, resulting in a significant reduction in the genetic base in many cultivated species [32]. This is evidenced by the findings of Viáfara-Vega et al. [67], who studied genetic diversity in cultivated lines of *C. annuum* and *C. frutescens* in Valle del Cauca, reporting low allelic diversity in cultivars due to high self-pollination and homozygosity, resulting in highly inbred lines. Selective sweeps and genetic bottlenecks have often led to the loss of other important traits [68], such as nutritional content or organoleptic properties. Therefore, our effort exploring ancient cultivars, landraces and wild relatives with greater phenotypic and genetic diversity promises recovering those lost [32,61], while unlocking the potential of the underutilized germplasm from northwest South America.

Capturing and preserving genetic diversity is not sufficient for improving current cultivars, as the factual utilization of the agrobiodiversity relies on pinpointing and mobilizing relevant alleles from exotic gene pools into improved, commercially accepted, varieties [69]. While germplasm banks play a strategic role in safeguarding valuable genetic variants that confer competitive advantages such as abiotic stress tolerance [70,71] or higher content of bioactive compounds [72], tools like genotyping-by-sequencing (GBS) and GWAS analyses allow for tracing novel alleles useful for fruit quality improvement [73].

In our particular case, the traits with the highest number of associated QTNs, namely fruit weight, capsaicin content, and soluble solids, explained up to 26.7%, 55.1%, and 8.3% of the phenotypic variation per associated QTN, respectively, in the reference genome. Similarly, in the pangenome, QTNs associated with consistency, luminosity, and capsaicin content accounted for up to 33.9%, 17.5%, and 57.4% of the variation per QTN, respectively. These traits and their associated QTNs emerged as strong candidates for improving fruit quality. The markers identified are suitable for introgressing exotic alleles from donor wild or landrace accessions into elite cultivars [74]. Moreover, the estimation of breeding values supports the development of genomic prediction models and the design of more efficient breeding strategies. This is especially relevant in countries with high levels of genetic diversity or unique and abundant biological reservoirs. In such contexts, these genomic prediction models facilitate the identification and selection of promising genetic materials for incorporation into breeding programs [75]. Altogether, the integration of molecular analyses with phenotypic characterization of germplasm, as demonstrated here, is a critical prerequisite for enhancing introgression-based improvement strategies in *Capsicum* [74].

Another major accomplishment of our work was the incorporation of pangenomic approaches (or variant graphs) alongside a linear reference genome, which together allows a broader understanding of genomic diversity, as it facilitates the detection of structural variants and alleles absent from conventional references [76]. In other species like tomato, the use of a pangenome significantly increased captured genetic variation, and the inclusion of structural variants raised trait heritability estimates by about 24% (from 0.33 to 0.41) compared to using a linear genome [77]. These values compare with our 43.1% increase in the explained variance for fruit quality traits when using the pangenomic approach. The improved resolution provided by the pangenome enhances the identification of genes related to quality traits, thereby increasing the potential for application in plant breeding programs aiming at unconventional breeding target for novel [77,78]. Leveraging this more representative genetic reservoir in terms of an expanded repertoire of germplasm and available reference alignments enables breeders to develop crops that are not only with added market value but also more resilient to climate change, while promoting sustainable agriculture aligned with global market demands and novel industrial applications [61,79].

## 4. Materials and Methods

### 4.1. Plant Material and Experimental Design

A total of 283 accessions from the germplasm bank at the AGROSAVIA Research station La Selva in Rionegro (province of Antioquia, Colombia) were evaluated, representing the five domesticated *Capsicum* species. This collection encompassed a wide genetic diversity of the genus in a key agrobiodiversity hotspot in northwest South America, likely missed by previous studies due to limited access to the local genetic resources.

Plants were cultivated in greenhouses on soil covered with agromulch, following a completely randomized block design and grouped by species, with a planting density of 45 cm between plants and 1.5 m between rows. Drip irrigation was applied for 15 min daily. Each row consisted of seven plants per genotype, and the fruits harvested from the five central plants were pooled for all subsequent analyses. Fruits were harvested at full physiological maturity based on genotype-specific criteria, ensuring average fruit weights above 1 g and total sample weights exceeding 200 g. Samples were stored at −20 °C after harvest in order to assess trait variation for 18 fruit traits, as detailed in the next sections.

### 4.2. Morphological Characterization of Fruit Traits

A total of four fruit morphological descriptors were evaluated based on guidelines from the International Plant Genetic Resources Institute [80]. A morphological trait matrix was generated from five fruits collected per plant across five plants per accession. Fruit weight was measured using an analytical balance, seed number was determined manually, and fruit width and length were measured with digital calipers.

### 4.3. Physicochemical Characterization

As part of the sample pretreatment for the measure of ten physicochemical fruit traits, fruits were disinfected using 0.02% sodium hypochlorite from Merck (Darmstadt, Germany) for 3 min, dried, and processed into a homogeneous paste (including seeds) for analysis.

For pH and Brix measures, each paste sample was analyzed in quintuplicate per accession. pH was measured directly using a calibrated ST3100 pH meter (OHAUS Latinoamerica, Ciudad de México, Mexico), and soluble solids were quantified using a digital refractometer (Milwaukee Instruments, Inc., Rocky Mount, NC, United States) on juice extracted from the fruit paste.

In terms of colorimetry, color parameters were quantified using a ColorFlex EZ spectrophotometer (Hunter Associates Laboratory, Resto, VA, USA) and EasyMatchQC software (v.4.70). CIELab variables recorded included lightness (L), chroma (C), and hue angle (h) [81].

The texture parameters included consistency, firmness, cohesiveness, and work of cohesion were evaluated with a TA.XTplus texture analyzer (Stable Micro Systems Ltd., UK) equipped with a 35 mm back extrusion accessory and Exponent software (v.6.1.16.0), using the “Ketchup back extrusion—KCH1_BEC” protocol. Test conditions were pre-test speed of 1 mm/s, test speed of 5 mm/s, post-test speed of 10 mm/s, target mode distance of 20 mm, and trigger force recorded in grams [82].

Finally, total solids and moisture content were determined via weight loss after drying [83]. Two grams of paste were dried at 105 °C for 2 h in aluminum trays using a forced-air Venticell convection oven (MMM Medcenter Einrichtungen GmbH, Planegg/München, Germany). Dry matter was calculated based on sample weight loss.

### 4.4. Chemical Characterization

As part of the sample pretreatment for the assessment of four chemical fruit traits, frozen pastes were thawed, and 50 g of sample were dried at 50 °C for 36 h in a convection oven. Dried samples were ground into a fine powder [84].

The quantification of capsaicinoids followed the method by Penagos et al. [85], in which 200 mg of powdered sample were extracted with 2 mL of HPLC-grade methanol from Merck (Darmstadt, Germany) by vortexing (3 min) and ultrasonication (30 min, 45 °C). The extract was processed by solid-phase extraction (Oasis HLB cartridges, 30 g/1 cc, Waters Corporation, Milford, MA, USA), and eluates were stored at −20 °C until analysis [14].

A calibration curve was generated with ultrapure standards of capsaicin and dihydrocapsaicin (Sigma-Aldrich; purities: 90% and 95%, respectively; concentrations: 50–500 mg/L). Analytes were detected via HPLC (Merck Hitachi, Lachrom Ultra model, Tokyo, Japan) under the following conditions: mobile phase: Type I water + 0.1% formic acid from Sigma-Aldrich (St. Lous, MO, United States) and HPLC-grade methanol (35:65); flow rate: 0.8 mL/min; wavelength: 280 nm; injection volume: 10 µL; temperature: 40 °C; column: Luna C18 (5 µm, 150 mm × 4.6 mm, Phenomenex Inc., Torrance, CA, USA) [14,85].

### 4.5. Quantification of Total Carotenoids

Depending on color intensity, 50–300 mg of powdered sample was extracted using ethyl acetate–dichloromethane (8:2 *v*/*v*) (Sigma-Aldrich) with 0.01% from Anmol chemicals (New York, United States), under low light, via vortexing and sonication. Extracts were dried and reconstituted in methanol. A 200 µL aliquot was transferred to 96-well plates and read at 450 nm using a Synergy H1 plate reader. An internal reference based on *Capsicum chinense* oleoresin was used. The calibration curve ranged from 15 to 85 mg/L [14].

### 4.6. Descriptive and Multivariate Analysis of Phenotypic Traits

All 18 fruit phenotypic variables were analyzed using the R software (v.4.4.2) [86]. To assess intra-genotype variability, data collection was carried out with replicates, and results were expressed as mean ± standard deviation (SD). Each trait was normalized using Tukey’s Ladder of Powers automatic transformation, employing the scale () function from R’s base package [86], especially for the GWAS approach based on mixed linear models (MLMs), which requires normalized quantitative data. Additionally, a Spearman correlation analysis was conducted using the corrplot package [87] to determine relationships between different variables.

### 4.7. DNA Extraction and Genotyping by Sequencing

For genomic DNA extraction, 20 mg of leaf tissue from one plant per accession (283 in total) was used. The extraction followed an in-house protocol optimized by AGROSAVIA’s lab (Mosquera, Colombia). Tissue was macerated in liquid nitrogen, with chloroform and phenol used during extraction, and isopropanol and 75% ethanol used for precipitation and purification. DNA was resuspended in 40 µL of molecular-grade ultrapure water, with 1 µL of RNase added. DNA concentration was measured by spectrophotometry using the NanoDrop 2000 (Thermo Fisher Scientific, Waltham, MA, USA) and by fluorometry using the Qubit dsDNA HS assay (Life Technologies, Ängelholm, Sweden).

For library preparation, the restriction enzyme *ApeK1* was used for DNA digestion, and the NEBNext^®^ Ultra™ II kit of Illumina^®^ (New England Biolabs, Ipswich, Massachusetts, USA) was used in single-index format for single-end reads. Libraries were quantified using the Qubit fluorometer and the Qubit™ dsDNA HS kit (Invitrogen, Waltham, Massachusetts, USA), and fragment size was determined using the TapeStation 4200 with the Agilent High Sensitivity D1000 ScreenTape kit (Applied Biosystems, Woburn, Massachusetts, USA). Libraries were temporally stored at −20 °C. Genomic data was obtained using genotyping-by-sequencing (GBS) [88].

### 4.8. Sequence Processing, Alignment, and SNP Calling

DNA sequencing was performed on the Illumina HiSeqXten platform (Macrogen, Seoul, South Korea) using single-end reads. Raw data were subjected to quality control using FastQC (of Illumina 1.9 encoding) [89]. Adapter sequences and low-quality regions were trimmed using Trimmomatic with the following parameters: ILLUMINACLIP:TruSeq3-PE:2:30:10, SLIDINGWINDOW:4:20, MINLEN:40, HEADCROP:12, LEADING:3, and TRAILING:3 [90]. For the pangenome graph mapping, cleaned reads were aligned using the Giraffe mapper with default parameters [91]. Quality was reassessed using FastQC to ensure Phred quality scores above 30 across all bases.

SNP calling was conducted using an automated variant-calling script available at https://github.com/FelipeLopez2019/Mr_Capsicum_SNP_Calling (accessed on 30 Nov 2024), applying the HaplotypeCaller function from the GATK4 pipeline [92] and BWA-mem for alignment [93]. The “MR *Capsicum*” protocol, standardized on a Linux server at Universidad Nacional (UNAL)’s Genetics Institute in Bogotá (Colombia), involved five key steps: (1) indexing the reference genome or pangenome, (2) generating individual gVCFs per sample, (3) merging gVCFs into a single file, (4) converting the final gVCF to a VCF, and (5) filtering the final VCF. The reference genome used was *Capsicum annuum* GCA_002878395.3 (Genome assembly UCD10Xv1.1), downloaded from NCBI (3.2 Gb, ~56X coverage, University of California, Davis) [25]. The pangenome was constructed following Liu et al. [17] using *vg toolkit* (v1.46.0) [94], mapping reads and generating genotype files. The pangenome graph used the *C. annuum* var. *annuum* Zhangshugang genome as the backbone and included polymorphism matrices derived from mapping *C. baccatum* var. *pendulum* PI 632928 and *C. pubescens* Grif 1614 reads [17]. Alignment statistics were obtained using the *flagstat* function of SAMtools 1.9 [95] on the Galaxy platform v.24.1.3 [96]. The resulting SNP matrix was filtered using TASSEL 5.2.94 [97], applying a minimum read depth threshold of 3X, a missing data rate ≤ 0.3 per locus and per sample, and a minor allele frequency (*maf*) ≥ 0.05, according to standard GWAS settings.

### 4.9. Population Structure and Kinship Analysis

To minimize false positives in each GWAS model, demographic fixed and random effects were accounted for. For fixed effects, population stratification was summarized through molecular principal component analysis (PCA), supervised analysis to verify the correspondence between genetic structure and taxonomic classification, and unsupervised clustering in the R packages *adegenet* [98], *factoextra* [99] and *NbClust* (v.4.4.1) [100].

In addition, individual ancestry coefficients were estimated using a sparse non-negative matrix factorization algorithm via the *sNMF* function of the LEA package in R, with 10 repetitions. The number of ancestral populations (*K*) was determined based on the cross-entropy criterion [101]. Finally, for random effects, kinship relationships were considered by computing the kinship matrix using the VanRaden algorithm implemented in the GAPIT package in R [102].

### 4.10. Identification of Loci Associated with Phenotypic Traits

Association analyses were conducted using the GAPIT package in R, applying the FarmCPU and BLINK algorithms to the 18 measured fruit phenotypic variables [26,27,102] for a total of 36 models per each reference genome and the pangenome alignment (72 models overall), and each model considered the fixed and random demographic effects gathered, as indicated in the previous section. Highly significant associations were identified using a strict Bonferroni-adjusted significance threshold with α = 0.05, which was based on the total number of SNPs evaluated and led to an effective threshold of −log10 = 6.14 or *p*-value = 7.30 × 10^−7^ (= 0.05/68481) for the reference genome and −log10 = 7.75 or *p*-value = 1.79 × 10^−6^ (= 0.05/27906) for the pangenome [103]. This criterion was applied across all GWAS models to define the quantitative trait nucleotides (QTNs) associated with key fruit quality traits. The results were visualized by expressing the significance threshold (α/total SNPs) on a logarithmic scale (−log10), using a customized R script to generate Q-Q and Manhattan plots.

### 4.11. Identification of Candidate Genes and Functional Annotation

An R script available at https://speciationgenomics.github.io/candidate_genes/ (accessed on 25 March 2025) was used to identify and annotate candidate genes following Ravinet & Meier [104]. Using the General Feature Format (GFF) annotation files of the reference genome *Capsicum annuum* GCA_002878395.3 and the pangenome genome *C. annuum* var. *annuum* Zhangshugang, gene coordinates were extracted and the genomic midpoint for each gene was stored. A sliding window of 50 kb, a conservative window size based on values reported in the literature [15], was then applied around the genomic positions of the candidate QTNs to identify genes located within this interval.

Additionally, an orthology analysis was performed using Orthofinder (v2.5.4) [105] to cluster orthologous genes across different *C. annuum* genome versions. Coding sequences (CDS) from *C. annuum* cv. CM334 [106], *C. annuum* Zunla-1 [107], and *C. annuum* G1-36576 [29], previously used to localize genes involved in capsaicinoid biosynthesis, were used along with CDS from the annotated reference genome in NCBI and the studied genomes. Once orthogroups (clusters of closely related orthologs and paralogs) were defined, the annotation of candidate genes flanking QTNs associated with capsaicin, dihydrocapsaicin, and total capsaicinoids was carried out.

In order to annotate candidate genes, orthogroups (HOGs) containing QTN-related genes were identified and the RefSeq-RNA accession codes for the genes in these HOGs were extracted. These codes were used to retrieve gene annotations from the DAVID bioinformatics server (https://davidbioinformatics.nih.gov/tools.jsp—accessed on 18 May 2025). Second, a personal database of genes involved in pungency and capsaicin biosynthesis, previously validated in the literature [15,29,107,108,109], was used to identify orthogroups containing a priori known capsaicinoid biosynthesis genes. Finally, the overlap between HOGs containing QTN-related genes and known capsaicin biosynthetic genes was evaluated. In this regard, one-to-one orthologs were identified between the NCBI reference genome and the pangenome, enabling the comparison of QTNs associated with the same trait across both assemblies within syntenic genomic regions.

## 5. Conclusions

The *Capsicum* germplasm from northwest South America exhibits remarkable fruit quality phenotypic and genomics diversity. Furthermore, the implementation of GWAS analyses using both a reference genome and a pangenome enabled the identification of SNPs associated with fruit quality traits, highlighting the utility of the pangenome in capturing additional structural variations. The joint characterization of morphological, physicochemical and chemical fruit traits, together with robust genomic screening, proved to be an effective strategy for identifying candidate QTNs with potential to assist the Introgression into elite lines of exotic variation for fruit quality. Ultimately, this study lays the groundwork for harnessing local underutilized germplasm in the development of improved varieties aimed at meeting growing demands in the natural, functional, and sustainable products market.

## Figures and Tables

**Figure 1 ijms-26-08205-f001:**
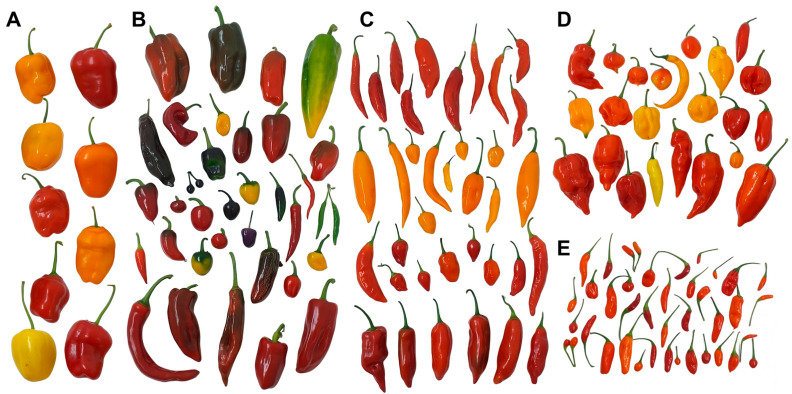
Representative fruit diversity from the Colombian *Capsicum* germplasm collection conserved by Agrosavia. (**A**) *C. pubescens*. (**B**) *C. annuum*. (**C**) *C. baccatum*. (**D**) *C. chinense*. (**E**) *C. frutescens*. Photographs taken from genotypes cultivated by Agrosavia in Rionegro, Antioquia.

**Figure 2 ijms-26-08205-f002:**
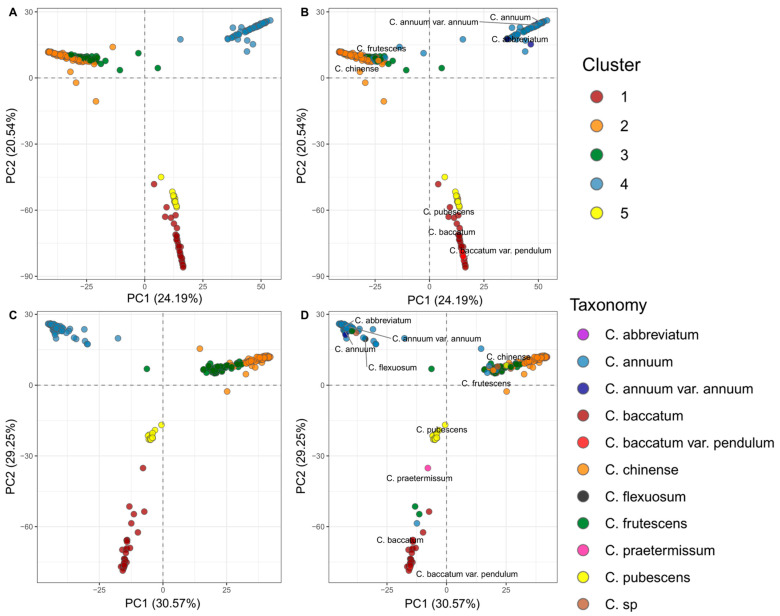
Genetic structure of *Capsicum* accessions based on reference and pangenome alignments. (**A**,**B**) Structure of 235 accessions using the reference genome: (**A**) unsupervised K—means clustering (K = 5); (**B**) clustering based on taxonomic classification. (**C**,**D**) Structure of 244 accessions using the pangenome: (**C**) unsupervised K—means clustering (K = 5); (**D**) clustering based on taxonomic classification. Colors in panels A and C indicate clusters: red, cluster 1; orange, cluster 2; green, cluster 3; light blue, cluster 4; yellow, cluster 5. Colors in B and D indicate taxonomy: purple, *C. abbreviatum*; light blue, *C. annuum*; dark blue, *C. annuum var. annuum*; dark red, *C. baccatum*; red, *C. baccatum var. pendulum*; orange, *C. chinense*; green, *C. frutescens*; yellow, *C. pubescens*; light brown, *C.* sp.

**Figure 3 ijms-26-08205-f003:**
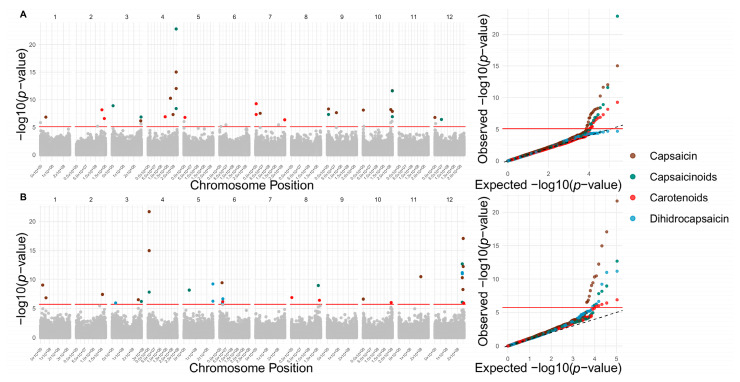
Genomic architecture of capsaicin (brown), capsaicinoids (Dark Teal), carotenoids (red), and dihydrocapsaicin (light blue). Manhattan plots (left panels) and Q–Q plots (right panels) from genome-wide association studies (GWAS) for the traits capsaicin, capsaicinoids, carotenoids, and dihydrocapsaicin (colored points). (**A**) Significant SNPs identified using the reference genome. (**B**) Significant SNPs identified using the pangenome.

**Figure 4 ijms-26-08205-f004:**
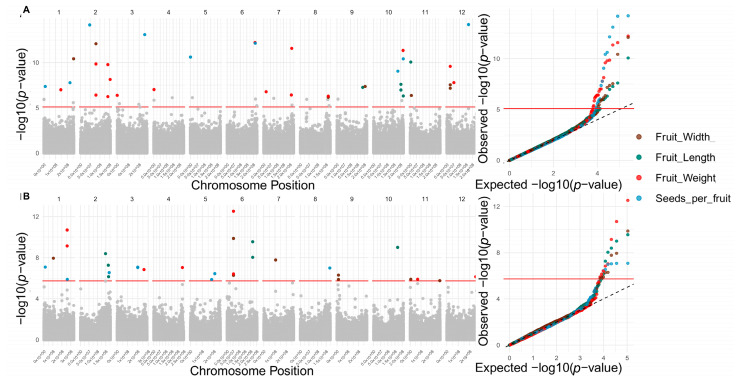
Genomic architecture of fruit morphology traits: width (brown), length (Dark Teal), fruit weight (red), and seed number per fruit (light blue). Manhattan plots (left panels) and Q–Q plots (right panels) from genome-wide association studies (GWAS) for each trait (colored points). (**A**) Candidate SNPs associated using the reference genome. (**B**) Candidate SNPs associated using the pangenome.

**Figure 5 ijms-26-08205-f005:**
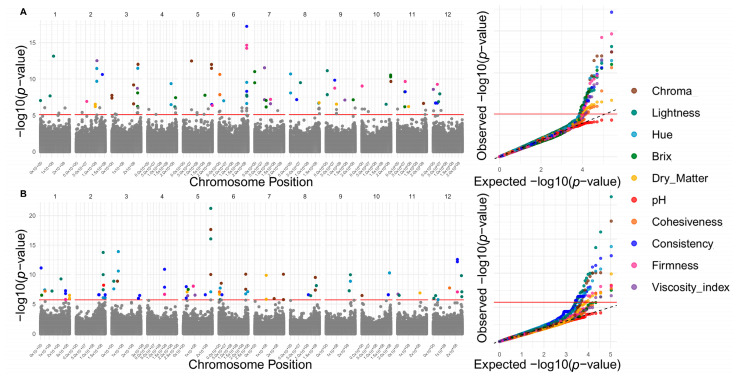
Genomic architecture of physicochemical traits: dry matter (mustard Yellow), Brix (dark green), pH (red), chroma (brown), lightness (dark Teal), hue (light blue), consistency (blue), firmness (pink), viscosity index (deep Violet), and cohesiveness (orange). Manhattan plots (left panels) and Q–Q plots (right panels) from GWAS for each trait (colored points). (**A**) Candidate SNPs associated using the reference genome. (**B**) Candidate SNPs associated using the pangenome.

**Figure 6 ijms-26-08205-f006:**
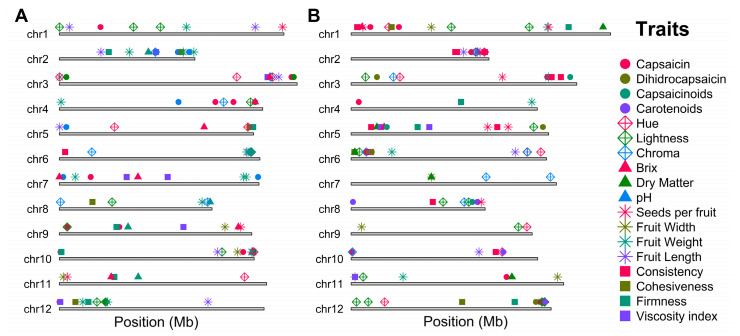
Multi-trait genomic architecture. Ideogram based on (**A**) the reference genome and (**B**) the pangenome. Shape and colors of the Traits: Capsaicin: Filled Circle-fuchsia, Dihidrocapsaicin: Filled Circle-olive green, Capsaicinoids: Filled Circle-deep green teal, Carotenoids: Filled Circle-purple; Hue: Diamon Cross- fuchsia, Lightness: Diamon Cross-olive green, Chroma: Diamon Cross-deep green teal; Brix: filled triangle-fuchsia, Dry Matter: filled triangle-olive green, pH: filled triangle-deep green teal; Seeds per fruit: Star-fuchsia, Fruit Width: Star-olive green, Fruit Weight: Star-deep green teal, Fruit Length: Star-purple; Consistency: filled square-fuchsia, Cohesiveness: filled square-olive green, Firmness: filled square-deep green teal, Viscosity index: filled square-purple.

**Figure 7 ijms-26-08205-f007:**
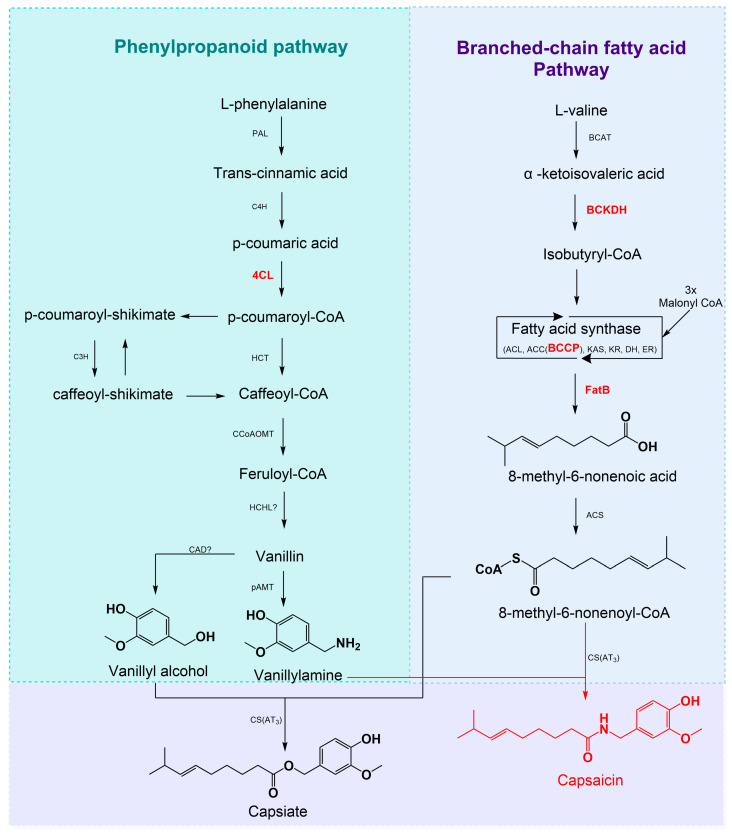
Diagram of capsaicinoid biosynthesis pathways and key genes. PAL, phenylalanine ammonia-lyase; C4H, cinnamate 4-hydroxylase; 4CL, 4-coumarate: CoA ligase; HCT, hydroxycinnamoyl transferase; CCoAOMT, caffeoyl-CoA 3-*O*-methyltransferase; C3H, coumarate 3-hydroxylase; HCHL, hydroxyl cinnamyl-CoA hydrase/lyase; AMT, aminotransferase; BCAT, branched chain amino acid aminotransferase; BCKDH, branched-chain α-ketoacid dehydrogenase; KAS, β-ketoacyl-ACP synthase; KR, β-ketoacyl-ACP reductase; DH, β-hidoxyacyl-ACP dehydratase; ER, enoyl-ACP reductase; ACL, acyl carrier protein; BCCP, biotin carboxyl carrier; ACC, acetyl-CoA carboxylase complex; FatB, acyl-ACP-thiesterase B; ACS, acetyl-CoA synthetase; and CS, capsaicin synthase. Adapted from [28,29].

## Data Availability

The raw data supporting the conclusions of this article will be made available by the authors without undue reservation. In addition, the BioSamples are hosted under the BioProject number PRJNA1309763 at NCBI.

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
