# Peer review of "Pangenomic and Phenotypic Characterization of Colombian Capsicum Germplasm Reveals the Genetic Basis of Fruit Quality Traits"

_ijms, 2025, doi:10.3390/ijms26178205_

Round 1
Reviewer 1 Report
Comments and Suggestions for Authors
The authors evaluated the phenotypic and genetic diversity of 283 accessions from the Colombian germplasm, and performed the GWAS analyses to identify QTNs associated the key fruit quality traits. These results prove important insights into genetic basis of traits in Capsicum. The manuscript could be accepted after addressing the following comments/suggestions:
Comments/suggestions:
- Introduction: They have talked a lot about the importance of Capsicum. I don't think all the information is necessary here. Just one or two sentences to summary the first three paragraphs are enough. The introduction should include more information about research status of QTLs/genes related to key fruit quality traits.
- L147-150: The references of reference genome and pangenome should be cited here.
- Line 146: The word "GBS-derived reads" is confused and there is no information about high-throughput sequencing, such as the how much data is produced from each sample, how about the results of their quality assessment.
- The abstract mentioned that genetic diversity of totally 283 accessions wasevaluated. However, only 235 accessions were used while mapping to reference genome, and 244 accessions were used while mapping to pangenome. The filtering criteria should be included in the manuscriptand the sample number should be corrected in abstract. Confusingly, data filter is usually performed before mapping to reference genome, why the numbers of accessions were different here between reference genome and pangenome? If the input is different, it makes no sense to do the comparative analysis of reference genome and pangenome.
- Line 263: How to determine the overlap QTNs between reference genome and pan-genome? Just based on their positions, which seemsnot comparable? The selected reference genome should be mapped to the pangenome, and then find out the overlap QTNs according to their matchable positions.
- Materials and Methods: "2.1" - "2.8" should be "4.1" - "4.8".
Author Response
For research article
Pangenomic and Phenotypic Characterization of Colombian Capsicum Germplasm Reveals the Genetic Basis of Fruit Quality Traits 
Summary
Thank you very much for taking the time to review this manuscript. Please find the detailed responses below and the corresponding revisions/corrections highlighted/in track changes in the re-submitted files.
Response to Reviewer 1 Comments
1. Introduction: They have talked a lot about the importance of Capsicum. I don't think all the information is necessary here. Just one or two sentences to summary the first three paragraphs are enough. The introduction should include more information about research status of QTLs/genes related to key fruit quality traits.
Thank you for your constructive comment. We agree that the introduction can be more concise and focused. In response, we have reduced the first two paragraphs into a single, more succinct paragraph that briefly summarizes the economic and nutritional relevance of Capsicum species:
Line 43-48: “The genus Capsicum ranks among the most economically important crops within the Solanaceae family. Between 2013 and 2023, the production of both chili and sweet peppers grew substantially, with dried forms increasing by 63.0% and fresh forms increasing by 22.4% [1]. This growth reflects rising demand for functional, nutrient-rich foods and the expanding use of Capsicum species in nutraceutical, pharmaceutical, and industrial ap-plications due to their high content of bioactive compounds [2,3].”
Additionally, we have expanded the latter part of the introduction to provide a more comprehensive overview of the current research status regarding QTLs and genes associated with key fruit quality traits:
Line 92-103: “In recent years, several studies have focused on identifying genomic regions associated with fruit quality traits in Capsicum using GWAS and QTL approaches. For instance, Liu et al. [17] constructed a graph-based pangenome from 500 wild and cultivated accessions, revealing distinct domestication sweeps between C. annuum and C. baccatum, as well as introgressions in C. chinense and C. frutescens. Likewise, López-Moreno et al. [18] and Fu et al. [19] uncovered multiple QTL and SNPs associated with key agronomic traits, confirming the complex and pleiotropic architecture of fruit domestication. Complementary studies have identified high-effect loci controlling capsaicinoid content [20,21], carotenoid pathways [22,23], primary metabolites [24], and fruit morphology [22], highlighting promising candidate genes such as RGLG1, HDG11, ARF23, and a UDP-glycosyltransferase. These findings collectively reinforce the utility of high-resolution genotyping in uncovering the complex genetic basis of fruit quality in Capsicum.”
2. L147-150: The references of reference genome and pangenome should be cited here.
Thank you for pointing this out. We agree with the reviewer that appropriate citations for the reference genome and the Capsicum pangenome were missing. Accordingly, we have revised the text in Lines 160-163 to include specific references that support the use of the reference genome and pangenome.
Lines 160-163: “The resulting reads from GBS were aligned to two genomic assemblies: the C. annuum reference genome (GCA_002878395.3) [25] and a pangenome assembled from C. annuum var. annuum Zhangshugang, C. baccatum var. pendulum PI 632928, and C. pubescens Grif 1614 [17].”
3. Line 150: The word "GBS-derived reads" is confused and there is no information about high-throughput sequencing, such as the how much data is produced from each sample, how about the results of their quality assessment.
Thank you for your comment. To clarify this aspect, we have expanded the sequencing description to include detailed metrics on library preparation, sequencing output, and quality assessment as indicated in Lines 152–158. Corresponding statistics are now provided in Table S4, and Line 160 has been revised to clearly introduce the alignment step.
Lines 152-158: “Library construction for genotyping-by-sequencing (GBS) yielded libraries with a mean insert size of 323.02 bp (SD = 32.58) and an average concentration of 173.62 nM (SD = 91.97), as detailed in Table S4. All libraries satisfied the minimum sequencing depth threshold of 5X, ensuring adequate coverage for downstream variant discovery. Quality metrics indicated high consistency across samples, with an average duplication rate of 35.59% (SD = 16.10%), a mean GC content of 41.09% (SD = 3.45%), and an average of 3,403,533 (SD = 1,807,881) sequencing reads per sample.”
Line 160: “The resulting reads from GBS were aligned to two genomic assemblies”
4. The abstract mentioned that genetic diversity of totally 283 accessions was evaluated. However, only 235 accessions were used while mapping to reference genome, and 244 accessions were used while mapping to pangenome. The filtering criteria should be included in the manuscriptand the sample number should be corrected in abstract. Confusingly, data filter is usually performed before mapping to reference genome, why the numbers of accessions were different here between reference genome and pangenome? If the input is different, it makes no sense to do the comparative analysis of reference genome and pangenome.
Thank you for this important observation. We understand the confusion and have clarified this point in the revised manuscript. The original dataset consisted of 283 accessions. However, due to differences in mapping efficiency and SNP calling pipelines for each genomic reference, the final number of accessions retained after quality filtering varied between the reference genome and the pangenome. Specifically, after applying standard quality control filters (e.g., missing data threshold, minor allele frequency), 235 accessions were retained in the reference genome-based dataset and 244 in the pangenome-based dataset.
The correction has been made in Lines 163 -165, which now reads:
“Starting from an initial set of 283 accessions, filtering steps retained 68,481 SNPs across 235 accessions in the reference-based dataset and 27,906 SNPs across 244 accessions in the pangenome-based dataset.”
5. Line 263: How to determine the overlap QTNs between reference genome and pan-genome? Just based on their positions, which seemsnot comparable? The selected reference genome should be mapped to the pangenome, and then find out the overlap QTNs according to their matchable positions.
Thank you for raising this important point. We agree that a direct comparison of QTNs based solely on genomic coordinates would be inappropriate given the structural differences between the reference genome and the pangenome.
To address this, we used an orthology based approach. Specifically, we first identified one to one orthologs between the NCBI reference genome and the genome assembly used to construct the pangenome. This enabled a biologically meaningful comparison of QTNs associated with the same trait across both genomic contexts. We then verified that these orthologous genes were located within similar genomic regions, supporting the comparability of the QTNs despite structural variation between assemblies.
This clarification has been added to the revised manuscript in Lines 722–725:
“In this regard, one-to-one orthologs were identified between the NCBI reference genome and the pangenome, enabling the comparison of QTNs associated with the same trait across both assemblies within syntenic genomic regions.”
6. Materials and Methods: "2.1" - "2.8" should be "4.1" - "4.8".
Thank you for pointing this out. We agree with the observation and have corrected the section numbering in the Materials and Methods part of the manuscript. The subheadings originally labeled as “2.1” through “2.8” have been updated to “4.1” through “4.8” to ensure consistency with the overall manuscript structure. These changes have been implemented in Lines 521–695 of the revised version.
Reviewer 2 Report
Comments and Suggestions for Authors
This manuscript investigates the genetic basis of fruit quality traits in Colombian Capsicum germplasm using GWAS and pangenomic approaches. The integration of diverse phenotypic data with SNP-based analyses across five domesticated species represents a valuable contribution to crop genetics and breeding. However, the manuscript requires major revision due to several critical issues. Key methodological details are unclear, important supplementary materials are missing, figures are difficult to read, and there is no experimental validation of candidate genes.
The abstract effectively communicates the aims of the study, the methodological basis, and the overall results. It accurately reflects the breadth of phenotypic and genomic analyses performed. However, some terms, such as “FarmCPU” and “BLINK”, require very brief explanation.
Supplementary Material: Although the authors reference several additional figures and tables (e.g., Figure S1, Tables S1–S3), these materials are not presented in the manuscript.
Line 107: Corrected “byidentifying”.
Figures 2–6: The font size of the axis legends, legends, and annotations is too small.
Lines 138–143: The results of the correlation matrix analysis are robust, but no explanation is given for the absence of some expected correlations (e.g., pH and dry matter content).
Lines 222–230: The discussion of large and small effect loci should be supplemented with information on the statistical thresholds for classifying “large” effects.
Although candidate genes were identified thoroughly, the study lacks experimental validation (e.g., gene expression analysis or functional studies) to confirm their role in the measured traits.
Lines 468–476: The statement about explaining phenotypic variance is substantial and suggests high heritability of the traits. However, it is necessary to clarify whether this variance is due to single or cumulative QTNs.
For traits with large effect sizes, the potential breeding value should be discussed in more detail.
The authors note the low overlap of SNP associations between the reference genome and the pangenome, but the discussion section should include a discussion of structural variations or alignment artifacts.
Lines 508–509: Indicate whether all five central plants of each genotype were pooled or analyzed separately for subsequent trait measurements.
Lines 552–554: Briefly justify the use of specific drying conditions in terms of linkage stability.
Include information on the average read depth achieved after filtering across samples.
Lines 662–665: Although the Bonferroni correction is stringent, authors are encouraged to note its conservativeness and discuss whether any putative associations were considered in secondary analyses.
Author Response
For research article
Pangenomic and Phenotypic Characterization of Colombian Capsicum Germplasm Reveals the Genetic Basis of Fruit Quality Traits 
Summary
Thank you very much for taking the time to review this manuscript. Please find the detailed responses below and the corresponding revisions/corrections highlighted/in track changes in the re-submitted files.
Response to Reviewer 2 Comments
1. However, some terms, such as “FarmCPU” and “BLINK”, require very brief explanation.
Thank you for your helpful suggestion. In response, we have updated the manuscript to briefly clarify the terms FarmCPU and BLINK within the relevant section.
This clarification has been incorporated into Lines 185-191:
“FarmCPU (Fixed and Random Model Circulating Probability Unification) and BLINK (Bayesian-information and Linkage-disequilibrium Iteratively Nested Key-way)-based GWAS models identified 144 significant SNPs in the reference genome asso-ciated with 16 fruit quality traits and 150 in the pangenome for all 18 evaluated traits (Figures 3-5 and table S1-S2). These algorithms enhance QTN detection by controlling population structure and false positives; FarmCPU alternates fixed and random models, while BLINK employs a faster, fixed-model approach based on Bayesian criteria [26,27].”
2. Supplementary Material: Although the authors reference several additional figures and tables (e.g., Figure S1, Tables S1–S3), these materials are not presented in the manuscript.
Thank you for bringing this to our attention. We apologize for the oversight. In the revised submission, we have now included all referenced supplementary as separate files accompanying the manuscript. These materials provide detailed support for the results and analyses presented in the main text.
3. Line 107: Corrected “byidentifying”.
Thank you for noticing this typographical error. The mistake has been corrected in Line 111, where “byidentifying” now correctly appears as “by identifying.”
4. Figures 2–6: The font size of the axis legends, legends, and annotations is too small.
Thank you for this helpful observation. We have revised Figures 2–6 to increase the font size of axis titles, legends, and annotations to improve readability and overall visual clarity. The updated figures have been included in the revised manuscript.
5. Lines 138–143: The results of the correlation matrix analysis are robust, but no explanation is given for the absence of some expected correlations (e.g., pH and dry matter content).
Thank you for this insightful comment. We have addressed this point in Lines 145–149 of the revised manuscript. Specifically, we clarify that the absence of some expected correlations such as between pH and dry matter content may be due to the broad genetic diversity of the analyzed accessions, which includes individuals from different Capsicum species and varying degrees of domestication. This high inter- and intra-specific variability may reduce the strength or consistency of associations across the full dataset, thus affecting statistical significance.
Lines 145 -149: “Moreover, no significant correlations were observed between chemical and textural parameters, or between chemical and colorimetric variables with pH and total and soluble solids (Figure S1). This lack of association may be attributed to the broad genetic diversity within the population, reflecting high inter- and intra-specific variability.”
6. Lines 222–230: The discussion of large and small effect loci should be supplemented with information on the statistical thresholds for classifying “large” effects.
Thank you for this important observation. The statistical thresholds used to define large- and small-effect loci are now clearly specified in Supplementary Tables S1 and S2, where the full range of effect sizes and their corresponding p-values are provided.
7. Although candidate genes were identified thoroughly, the study lacks experimental validation (e.g., gene expression analysis or functional studies) to confirm their role in the measured traits.
Regarding candidate gene identification, we note that all reported loci were annotated based on previously characterized genes in the literature and established biological pathways associated with the evaluated traits. However, as now emphasized in Lines 446–448, these associations remain predictive and must be confirmed experimentally through functional validation approaches (e.g., gene expression profiling, genetic complementation, or genome editing) and replication in independent populations.
Lines 446-448: “Moreover, functional validation of significantly associated genes is essential, through genetic complementation and genome editing techniques, along with replication of associations in independent populations [61].”
8. Lines 468–476: The statement about explaining phenotypic variance is substantial and suggests high heritability of the traits. However, it is necessary to clarify whether this variance is due to single or cumulative QTNs.
Thank you for this important clarification. In the revised version, we now explicitly state that the percentage of phenotypic variance explained refers to the effect per QTN, not the cumulative effect of multiple loci. This clarification has been incorporated into Lines 489 and 491 of the manuscript to avoid misinterpretation. As detailed in Supplementary Tables S1 and S2, each QTN is reported individually with its associated effect size and variance explained.
Lines 487-492: “In our particular case, the traits with the highest number of associated QTNs, namely fruit weight, capsaicin content, and soluble solids, explained up to 26.7%, 55.1%, and 8.3% of the phenotypic variation per associated QTN, respectively, in the reference genome. Similarly, in the pangenome, QTNs associated with consistency, luminosity, and capsai-cin content accounted for up to 33.9%, 17.5%, and 57.4% of the variation per QTN, respectively.”
9. For traits with large effect sizes, the potential breeding value should be discussed in more detail.
Thank you for this helpful suggestion. In response, we have revised the manuscript to strengthen the discussion regarding the breeding value of the evaluated traits. This emphasis has been incorporated in Lines 494–499, highlighting how the estimation of breeding values can guide the implementation of genomic prediction and support the selection of promising genetic materials for crop improvement programs.
Lines 494 – 499: “Moreover, the estimation of breeding values supports the development of genomic prediction models and the design of more efficient breeding strategies. This is especially relevant in countries with high levels of genetic diversity or unique and abundant biological reservoirs. In such contexts, these models facilitate the identification and selection of promising genetic materials for incorporation into breeding programs[75].”
10. The authors note the low overlap of SNP associations between the reference genome and the pangenome, but the discussion section should include a discussion of structural variations or alignment artifacts.
Thank you for this relevant comment. As clarified in the corrected version of the manuscript, we did not use an alignment-based approach to compare QTNs between the pangenomic and reference genome approaches. Instead, we used an orthology-based approach, where we identified one-to-one orthologs associated with the same trait in the pangenome analysis and the reference genome analysis.
11. Lines 508–509: Indicate whether all five central plants of each genotype were pooled or analyzed separately for subsequent trait measurements.
Thank you for pointing this out. We have clarified this in the revised manuscript. As stated in Lines 532-533, the fruits harvested from the five central plants per genotype were pooled for all subsequent trait measurements.
Lines 532-533: “Each row consisted of seven plants per genotype, and the fruits harvested from the five central plants were pooled for all subsequent analyses.”
12. Lines 552–554: Briefly justify the use of specific drying conditions in terms of linkage stability.
Thank you for your comment. The drying conditions used in our protocol were selected based on a previously validated study, which demonstrated that the chosen temperature parameters ensure the stability of the target analytes.
The corresponding references cited to support the methodological choice:
84. Hailu G, Derbew B. Extent, Causes and Reduction Strategies of Postharvest Losses of Fresh Fruits and Vegetables – A Review. J Biol Agric Healthc [Internet]. 2015 [cited 2025 Feb 7];5:49–64. Available from: https://www.iiste.org/Journals/index.php/JBAH/article/view/20627
85. Penagos-Calvete D, Guauque-Medina J, Villegas-Torres MF, Montoya G. Analysis of triacylglycerides, carotenoids and capsaicinoids as disposable molecules from Capsicum agroindustry. Hortic Environ Biotechnol. 2019;60:227–38.
13. Include information on the average read depth achieved after filtering across samples.
Thank you for the suggestion. The manuscript already specifies the minimum read depth of 3X used during SNP filtering. This information is stated in Line 663.
Line 662-664: “The resulting SNP matrix was filtered using TASSEL 5.2.94 [97], applying a minimum read depth threshold of 3X, a missing data rate ≤ 0.3 per locus and per sample and a minor allele frequency (maf) ≥ 0.05, according to standard GWAS settings.”
14. Lines 662–665: Although the Bonferroni correction is stringent, authors are encouraged to note its conservativeness and discuss whether any putative associations were considered in secondary analyses.
Thank you for this thoughtful suggestion. In this study, we deliberately adopted the Bonferroni correction due to its stringency, as we prioritized controlling the false positive rate over the inclusion of potentially spurious associations. Given the complexity of the population and the number of traits analyzed, we chose to limit the identification of associations to those with the highest statistical support. While we acknowledge that this approach may overlook some true positives, we did not pursue additional secondary analyses of sub-threshold associations to avoid inflating the risk of false discoveries.

Round 2
Reviewer 2 Report
Comments and Suggestions for Authors
The authors provided informative responses to all questions and addressed all comments thoroughly. For future revisions, I recommend avoiding continuous text when replying to comments, as it can make the responses harder to read.
Author Response
We sincerely apologize for this oversight. We understand that continuous text can make responses more difficult to follow. For future revisions, we will ensure that our replies to reviewer comments are presented in a clear, separated, and well-structured format. Below, we provide the revised responses separated for easier reading.
For research article
Pangenomic and Phenotypic Characterization of Colombian Capsicum Germplasm Reveals the Genetic Basis of Fruit Quality Traits 
Summary
Thank you very much for taking the time to review this manuscript. Please find the detailed responses below and the corresponding revisions/corrections highlighted/in track changes in the re-submitted files.
Response to Reviewer 2 Comments
1. However, some terms, such as “FarmCPU” and “BLINK”, require very brief explanation.
Thank you for your helpful suggestion. In response, we have updated the manuscript to briefly clarify the terms FarmCPU and BLINK within the relevant section.
This clarification has been incorporated into Lines 185-191:
“FarmCPU (Fixed and Random Model Circulating Probability Unification) and BLINK (Bayesian-information and Linkage-disequilibrium Iteratively Nested Key-way)-based GWAS models identified 144 significant SNPs in the reference genome asso-ciated with 16 fruit quality traits and 150 in the pangenome for all 18 evaluated traits (Figures 3-5 and table S1-S2). These algorithms enhance QTN detection by controlling population structure and false positives; FarmCPU alternates fixed and random models, while BLINK employs a faster, fixed-model approach based on Bayesian criteria [26,27].”
2. Supplementary Material: Although the authors reference several additional figures and tables (e.g., Figure S1, Tables S1–S3), these materials are not presented in the manuscript.
Thank you for bringing this to our attention. We apologize for the oversight. In the revised submission, we have now included all referenced supplementary as separate files accompanying the manuscript. These materials provide detailed support for the results and analyses presented in the main text.
3. Line 107: Corrected “byidentifying”.
Thank you for noticing this typographical error. The mistake has been corrected in Line 111, where “byidentifying” now correctly appears as “by identifying.”
4. Figures 2–6: The font size of the axis legends, legends, and annotations is too small.
Thank you for this helpful observation. We have revised Figures 2–6 to increase the font size of axis titles, legends, and annotations to improve readability and overall visual clarity. The updated figures have been included in the revised manuscript.
5. Lines 138–143: The results of the correlation matrix analysis are robust, but no explanation is given for the absence of some expected correlations (e.g., pH and dry matter content).
Thank you for this insightful comment. We have addressed this point in Lines 145–149 of the revised manuscript. Specifically, we clarify that the absence of some expected correlations such as between pH and dry matter content may be due to the broad genetic diversity of the analyzed accessions, which includes individuals from different Capsicum species and varying degrees of domestication. This high inter- and intra-specific variability may reduce the strength or consistency of associations across the full dataset, thus affecting statistical significance.
Lines 145 -149: “Moreover, no significant correlations were observed between chemical and textural parameters, or between chemical and colorimetric variables with pH and total and soluble solids (Figure S1). This lack of association may be attributed to the broad genetic diversity within the population, reflecting high inter- and intra-specific variability.”
6. Lines 222–230: The discussion of large and small effect loci should be supplemented with information on the statistical thresholds for classifying “large” effects.
Thank you for this important observation. The statistical thresholds used to define large- and small-effect loci are now clearly specified in Supplementary Tables S1 and S2, where the full range of effect sizes and their corresponding p-values are provided.
7. Although candidate genes were identified thoroughly, the study lacks experimental validation (e.g., gene expression analysis or functional studies) to confirm their role in the measured traits.
Regarding candidate gene identification, we note that all reported loci were annotated based on previously characterized genes in the literature and established biological pathways associated with the evaluated traits. However, as now emphasized in Lines 446–448, these associations remain predictive and must be confirmed experimentally through functional validation approaches (e.g., gene expression profiling, genetic complementation, or genome editing) and replication in independent populations.
Lines 446-448: “Moreover, functional validation of significantly associated genes is essential, through genetic complementation and genome editing techniques, along with replication of associations in independent populations [61].”
8. Lines 468–476: The statement about explaining phenotypic variance is substantial and suggests high heritability of the traits. However, it is necessary to clarify whether this variance is due to single or cumulative QTNs.
Thank you for this important clarification. In the revised version, we now explicitly state that the percentage of phenotypic variance explained refers to the effect per QTN, not the cumulative effect of multiple loci. This clarification has been incorporated into Lines 489 and 491 of the manuscript to avoid misinterpretation. As detailed in Supplementary Tables S1 and S2, each QTN is reported individually with its associated effect size and variance explained.
Lines 487-492: “In our particular case, the traits with the highest number of associated QTNs, namely fruit weight, capsaicin content, and soluble solids, explained up to 26.7%, 55.1%, and 8.3% of the phenotypic variation per associated QTN, respectively, in the reference genome. Similarly, in the pangenome, QTNs associated with consistency, luminosity, and capsai-cin content accounted for up to 33.9%, 17.5%, and 57.4% of the variation per QTN, respectively.”
9. For traits with large effect sizes, the potential breeding value should be discussed in more detail.
Thank you for this helpful suggestion. In response, we have revised the manuscript to strengthen the discussion regarding the breeding value of the evaluated traits. This emphasis has been incorporated in Lines 494–499, highlighting how the estimation of breeding values can guide the implementation of genomic prediction and support the selection of promising genetic materials for crop improvement programs.
Lines 494 – 499: “Moreover, the estimation of breeding values supports the development of genomic prediction models and the design of more efficient breeding strategies. This is especially relevant in countries with high levels of genetic diversity or unique and abundant biological reservoirs. In such contexts, these models facilitate the identification and selection of promising genetic materials for incorporation into breeding programs[75].”
10. The authors note the low overlap of SNP associations between the reference genome and the pangenome, but the discussion section should include a discussion of structural variations or alignment artifacts.
Thank you for this relevant comment. As clarified in the corrected version of the manuscript, we did not use an alignment-based approach to compare QTNs between the pangenomic and reference genome approaches. Instead, we used an orthology-based approach, where we identified one-to-one orthologs associated with the same trait in the pangenome analysis and the reference genome analysis.
11. Lines 508–509: Indicate whether all five central plants of each genotype were pooled or analyzed separately for subsequent trait measurements.
Thank you for pointing this out. We have clarified this in the revised manuscript. As stated in Lines 532-533, the fruits harvested from the five central plants per genotype were pooled for all subsequent trait measurements.
Lines 532-533: “Each row consisted of seven plants per genotype, and the fruits harvested from the five central plants were pooled for all subsequent analyses.”
12. Lines 552–554: Briefly justify the use of specific drying conditions in terms of linkage stability.
Thank you for your comment. The drying conditions used in our protocol were selected based on a previously validated study, which demonstrated that the chosen temperature parameters ensure the stability of the target analytes.
The corresponding references cited to support the methodological choice:
84. Hailu G, Derbew B. Extent, Causes and Reduction Strategies of Postharvest Losses of Fresh Fruits and Vegetables – A Review. J Biol Agric Healthc [Internet]. 2015 [cited 2025 Feb 7];5:49–64. Available from: https://www.iiste.org/Journals/index.php/JBAH/article/view/20627
85. Penagos-Calvete D, Guauque-Medina J, Villegas-Torres MF, Montoya G. Analysis of triacylglycerides, carotenoids and capsaicinoids as disposable molecules from Capsicum agroindustry. Hortic Environ Biotechnol. 2019;60:227–38.
13. Include information on the average read depth achieved after filtering across samples.
Thank you for the suggestion. The manuscript already specifies the minimum read depth of 3X used during SNP filtering. This information is stated in Line 663.
Line 662-664: “The resulting SNP matrix was filtered using TASSEL 5.2.94 [97], applying a minimum read depth threshold of 3X, a missing data rate ≤ 0.3 per locus and per sample and a minor allele frequency (maf) ≥ 0.05, according to standard GWAS settings.”
14. Lines 662–665: Although the Bonferroni correction is stringent, authors are encouraged to note its conservativeness and discuss whether any putative associations were considered in secondary analyses.
Thank you for this thoughtful suggestion. In this study, we deliberately adopted the Bonferroni correction due to its stringency, as we prioritized controlling the false positive rate over the inclusion of potentially spurious associations. Given the complexity of the population and the number of traits analyzed, we chose to limit the identification of associations to those with the highest statistical support. While we acknowledge that this approach may overlook some true positives, we did not pursue additional secondary analyses of sub-threshold associations to avoid inflating the risk of false discoveries.
